
# Climate Impact of Finnish Air Pollutants and Greenhouse Gases using Multiple Emission Metrics

Kaarle J. Kupiainen[1], Borgar Aamaas[2], Mikko Savolahti[1], Niko Karvosenoja[1], Ville-Veikko Paunu[1]

[1]Finnish Environment Institute, Mechelininkatu 34a, P.O. Box 140, FI-00251 Helsinki, Finland
[2]CICERO Center for International Climate Research, PB 1129 Blindern, 0318 Oslo, Norway

*Correspondence to*: Kaarle J Kupiainen (kaarle.kupiainen@ymparisto.fi or kaarle.kupiainen@ym.fi)

**Abstract.** We present a case study where emission metric values from different studies are applied to estimate global and Arctic temperature impacts of emissions from a Northern European country. This study assesses the climate impact of Finnish air pollutant and greenhouse gas emissions in 2000-2010 as well as future emissions until 2030. The pollutants
included are SO2, NOX, NH3, NMVOC, BC, OC and CO as well as CO2, CH4 and N2O, and our study is the first one for Finland to include all of them in one coherent dataset. These pollutants have different atmospheric lifetimes and influence the climate differently; hence, we look at different climate metrics and time horizons. The study uses the Global Warming Potential (GWP), the Global Temperature change Potential (GTP) and the Regional Temperature change Potential (RTP) with different time scales for estimating the climate impacts by species and sectors globally and in the Arctic. We compare
the climate impacts of emissions occurring in winter and summer. This assessment is an example of how the climate impact of emissions from small countries and sources can be estimated, as it is challenging to use climate models to study the climate effect of national policies in a multi-pollutant situation. Our methods are applicable to other countries and regions and present a practical tool to analyse the climate impacts in multiple dimensions, such as assessing different sectors and mitigation measures. While our study focuses on short-lived climate forcers, we find that the CO2 emissions have the most
significant climate impact, and the significance increases with longer time horizons. In the short term, emissions of especially CH4 and BC played an important role as well. The warming impact of BC emissions is enhanced during winter. There can be relatively large differences between results from studies using different metrics, which can partly be explained by different study setup and inherent uncertainty.

## 1. Introduction

The Paris Agreement and its target of "holding the increase in the global average temperature to well below 2° C above pre-industrial levels and pursuing efforts to limit the temperature increase to 1.5° C above pre-industrial levels" (UNFCCC, 2015) provides an important framework for individual countries to consider the climate impacts and mitigation possibilities of its emissions. Globally CO2 and greenhouse gas emissions are key components in achieving the targets of the agreement, but the role of short-lived climate forcers (SLCFs) should also be studied as additional drivers of the surface temperatures.



The climate effect of emission reductions of air pollutants, particularly black carbon and tropospheric ozone, have been a
focus of research in last few years (Shindell et al. 2012, Bond et al. 2013, Smith and Mizrahi, 2013, Stohl et al. 2015). Since
air pollutants can either cool or warm the climate on different timescales depending on the species, emission reduction
policies from a climate perspective have to be designed to take into account the net-effect of multiple pollutants
(UNEP/WMO 2011, Stohl et al. 2015). The pollutants considered to have most climate relevance are termed Short-lived

Climate Pollutants (SLCP) or Short-lived Climate Forcers (SLCF), depending on the context. SLCPs consist of the warming
components black carbon (BC), methane (CH4), ozone (O3), and sometimes also include HFC compounds. SLCFs include
the warming components, but also the ones that cool the climate, such as organic carbon (OC) and sulfate. Policies focusing
on warming SLCPs have indeed been suggested as supplements to greenhouse gas reductions (UNEP/WMO 2011, Shindell
et al. 2012, Rogelj et al. 2014, Stohl et al 2015, Shindell et al. 2017). Since in this study we are interested on both warming

and cooling effects of the air pollutants we use the term SLCFs.

Modelling studies by UNEP/WMO (2011) and Stohl et al. (2015) suggested that the climate response of SLCF mitigation is
strongest in the Arctic region. The Arctic region is of particular interest, since in the past 50 years the Arctic has been
warming twice as rapidly as the world as a whole, and has experienced significant changes in ice and snow covers as well as

permafrost (AMAP, 2017). AMAP (2011 and 2015) as well as Sand et al. (2016) demonstrated that emission reductions of
SLCFs in the Northern areas have the largest temperature response on the Arctic climate per unit of emissions reduced, with
the Nordic countries (Denmark, Finland, Iceland, Norway, and Sweden) and Russia having the largest impact when
compared with the other Arctic countries, Unites States and Canada.

Shindell et al. (2017); Ocko et al. (2017) have argued for assessing both near- and long-term effects of climate policy.
However, comparing the climate impacts of SLCFs, CO2, and other pollutants is not straightforward. Emission metrics is
one way of enabling a comparison as they provide a conversion rate between emissions of different species into a common
unit, for example CO2-equivalent emissions. Common emission metrics are the Global Warming Potential (GWP) (IPCC,
1990) and the Global Temperature change Potential (GTP) (Shine et al., 2005). The GWP compares the integrated radiative

forcing (RF) of a pulse emission of a given species relative to the integrated RF of a pulse emission of CO2. Since the
UNFCCC reporting procedure uses the GWP with a 100 yr time horizon (GWP100) as a reporting guideline, it has become
the most common metric to report greenhouse gas emissions. The GTP is an alternative to GWP and it compares the
temperature change at a point in time due to a pulse emission of a species relative to the temperature change of a pulse
emission of CO2. The GTP combines the changes in the radiative forcing induced by the different species with the

temperature response of the climate system and thus has been argued to relate better with climate effects (Shine et al., 2005).
Both GWP and GTP focus on the global response, while the temperature impact can also be analyzed on a regional scale, i.e.
the Arctic, applying Regional Temperature change Potential (RTP) (Shindell and Faluvegi, 2010). Even for an uniform
forcing, there will be spatial patterns in the temperature response. The metrics can be presented in absolute forms of radiative





forcing (AGWP) or temperature perturbation (AGTP, ARTP) as well as normalized to the response of CO2 (GTP, GWP,
RTP). Especially for short-lived species, the climate impact depends on the location and timing of the emissions, which is
reflected in the RTPs as well as in the global response for GTP and GWP. On a global scale, Unger et al. (2009) attributed
the RF to different economic sectors, while Aamaas et al. (2013) estimated the climate impact of different sectors based on
different emission metrics for global emissions, as well as regionally for the United States of America, China and the
European Union.


In this study we assess the climate impact of Finnish air pollutants (SO2, NOX, NH3, NMVOC, BC, OC and CO) and
greenhouse gas emissions (CO2, CH4 and N2O) in the past (2000-2010) and until 2030, according to a baseline emission
projection. We utilize emission metric values from several new studies relevant for Finland.

Finnish emissions and their climate response are relatively small compared with emissions from larger regions, let alone the
globe. Therefore it is challenging to use climate models to study the climate effect of national policies and to analyze the role
of each pollutant and sector. This study demonstrates a method to overcome this challenge by the use of emission metrics.
The method is applicable in other countries or regions as well and has been used in connection with the Norwegian work on
SLCPs (Norwegian Environment Agency, 2014) (Hodnebrog et al., 2014).


The Methodology section describes the construction and background data of the emission inventory and the future scenario
as well as the emission metrics used. In Results we describe the emissions and their climate impacts first focusing at the
historical emissions (2000-2010) and then at a future projection until 2030. We also discuss separately the regional
temperature effect of emissions on the Arctic and compare the results obtained with different metric studies. In the
Conclusions section we will summarize the main findings and conclude on the major scientific and policy relevant messages.

The objectives of this study were to (1) produce an integrated multi-pollutants emission dataset for Finland for 2000 to 2030,
(2) compare multiple climate metrics and assess their suitability for a Northern country like Finland, (3) estimate the climate
impact of Finnish air pollutants and greenhouse gases for the period 2000 to 2030 utilizing selected climate metrics, and (4)
suggest a set of global and regional climate metrics to be used in connection with Finnish SLCF emissions.

## 2. Methodology

Finland is one of the Nordic countries situated, between latitudes 60°N and 70°N. It has a population of 5.5 Million people
with an average population density 17.9 inhabitants per square kilometer (for comparison: EU average is 117 inhabitants per
square kilometer). Although much of the population is concentrated to the South of the country, the scarce population
compared with the size of the country makes transport of goods and people an important activity. The Northern location of



the country in turn results in a high demand for energy to heat households, and the economy is largely based on energy-intensive industry.

## 2.1 Emissions

The historical emissions of SO2, NOX, BC and OC in 2000, 2005 and 2010 are estimated based on the data in the Finnish
Regional Emission Scenario (FRES) model (Karvosenoja, 2008). Emissions of NH3, VOC, CO2, CH4 and N2O are from the national air pollutant and greenhouse gas emission inventories as reported to the UNFCCC and the UNECE Convention on Long-Range Transboundary Air Pollution (CLRTAP). The CO emission data is estimated with the GAINS model (http://gains.iiasa.ac.at; Amann et al. 2011). The data sources by pollutant are presented in Table 1. Emissions of CO2 are presented according to the IPCC guidelines, which assume biomass as carbon neutral. However, this definition is disputed
and e.g., Cherubini et al. (2011) present emission metric values that account for CO2 emissions from biomass. Although the historical emission data emanates from different data sources (Table 1), they have been checked for consistency and are based essentially on the same statistical sources. We aggregated the data and performed specific analyses for the following eight major economic sectors: energy production (ENE IND), industrial processes (PROC), road transport (TRA RD), off-road transport and machinery (TRA OT), domestic combustion (DOM), waste (WST), agriculture (AGR), and other
(OTHER).

Table 1. Data sources of the historical emission data for 2000-2010

The assumptions about the future energy use, transport and other activities in Finland follow Finland's 2013 National
Climate and Energy Strategy (Ministry of Employment and the Economy, 2013) and its' baseline scenario that fulfils the agreed EU targets and specific national targets for share of renewables and emission reductions in the non-ETS sector. Table 2 shows the primary energy consumption by fuel in Finland in 2010 and 2030. The 2013 National Climate and Energy Strategy assumes the future prevalence of wood heating to remain at 2011 level, which is estimated to lead to a decreased wood consumption, due to increasing energy efficiency in housing. The future emission projection was estimated with the
Finnish Regional Emission Scenario (FRES) model which used the activity estimates from the 2013 National Climate and Energy Strategy (Table 2) as a basis.

Table 2. Primary energy consumption in Finland (TWh a-1) (Ministry of Employment and the Economy, 2013)

## 2.2 Emission metrics

This work studies Finnish emissions with several climate metrics and focuses particularly on three of them, the Absolute Global Warming Potential (AGWP) (IPCC, 1990), Absolute Global Temperature change Potential (AGTP) (Shine et al.,



2005), and Absolute Regional Temperature change Potential (ARTP) (Shindell and Faluvegi, 2010). AGWP at time horizon H for emissions of pollutant i in emission season s from emission sector t is defined as


$$AGWP_{i,s,t}(H) = \int_0^H RF_{i,s,t}(t)dt \,, \qquad (1)$$

where RF is the time-varying radiative forcing given a unit mass pulse emission at time zero. Since two recent studies (Aamaas et al., 2016;Aamaas et al., 2017) have separated between emission during summer (May-October) and winter (November-April), we make this separation when possible. AGTP is given as


$$AGTP_{i,s,t}(H) = \int_0^H RF_{i,s,t}(t)IRF_T(H-t)dt \,. \qquad (2)$$

IRFT(H-t) is the temperature response, or impulse response function for temperature, at time H to a unit radiative forcing at time t. The ARTP is similar to AGTP, but gives the temperature response in latitude bands m:

$$ARTP_{i,m,s,t}(H) = \sum_l \int_0^H \frac{F_{l,i,s,t}(t)}{E_{i,s,t}} \times RCS_{i,s,l,m} \times R_T(H-t)dt, \qquad (3)$$

where Fl,i,s,t(t) is the radiative forcing in latitude band l and RCSi,l,m is matrix of unitless regional response coefficients based on the ARTP concept (Collins et al., 2013). In one of the papers (Sand et al., 2016), RCS differ for some of the different sectors, such as BC emissions in the Nordic countries from the domestic sector have about 15 percent higher sensitivity than BC emissions from energy and industry. Other papers do not provide this information on a sector level, and

we must use the same RCS for all emission sectors.

The ARTP method divides the world into four latitude bands: southern mid-high latitudes (90-28° S), the Tropics (28° S-28° N), northern mid-latitudes (28-60° N), and the Arctic (60-90° N). We will focus on the temperature response in the Arctic, as well as the global mean response.


Some of the studies separate the net response for a pollutant into various processes. For the aerosols, the radiative efficiencies often include the aerosol direct and 1st indirect (cloud-albedo) effect. In addition, BC deposition on snow and semi-direct effect may also be considered for BC. The ozone precursors build on the processes short-lived ozone effect, methane effect, and methane-induced ozone effect, as well as the aerosol direct and 1st indirect effects.


All these emission metrics (AGWP, AGTP, ARTP) can be normalized to the corresponding effect of CO2, where M is GWP, GTP, or RTP:



$$M_i(t) = \frac{AM_i(t)}{AM_{CO_2}(t)}.$$
(4)


The pollutants we include in our analysis ($SO_2$, $NO_X$, $NH_3$, NMVOC, BC, OC, CO, $CO_2$, $CH_4$, and $N_2O$) have very different atmospheric lifetimes and impact pathways. For the GHGs ($CO_2$, $CH_4$, and $N_2O$), we use the climate metric parameterization in IPCC AR5 (Myhre et al., 2013), but with an upward revision of 14% for $CH_4$ to account for the larger radiative forcing calculated by Etminan et al. (2016). The atmospheric decay of $CO_2$ is parameterized based on the Bern

Carbon Cycle Model (Joos et al., 2013) as reported in Myhre et al. (2013). We assume that the relative temperature response pattern in the four latitude bands is roughly the same for all the GHGs, and we base our calculations on the latitude pattern for $CH_4$ in Aamaas et al. (2017).

For all the other pollutants ($SO_2$, $NO_X$, $NH_3$, NMVOC, BC, OC, and CO), we use several recent studies that are relevant for

the emission location, Finland (Aamaas et al., 2016;Aamaas et al., 2017;Sand et al., 2016). The different studies are also compared, and we discuss how the choice of metric dataset influences the results. For the global temperature response, our main pick is the ARTP values published in Aamaas et al. (2017), while for the Arctic response, we compare ARTP values from Aamaas et al. (2017);Sand et al. (2016). These studies separate between different processes for each pollutant, such as the direct atmospheric RF and snow albedo effect of snow for BC. In our results, we include some discussion of these

different processes.

No studies have presented climate metric values specific for Finnish emissions. The default choice would be to use climate metric values based on global emissions, while we believe using smaller emission regions near or including Finland is more representative than applying the global average. The most relevant emission regions in the three selected studies are Europe

(consisting of the Western Europe, Eastern members of the European Union, and Turkey, up to 66°N) for Aamaas et al. (2016);Aamaas et al. (2017) and the Nordic countries for Sand et al. (2016). The Nordic countries is a smaller region and geographically more representative for Finland than Europe; however, Sand et al. (2016) provide only climate metric values for the Arctic response and only includes  BC, OC, and $SO_2$.

For all the pollutants, the IRF for temperature comes from the Hadley CM3 climate model (Boucher and Reddy, 2008). Hence, our temperature calculations are based on a climate sensitivity of 3.9 K warming for a doubling in $CO_2$ concentration. We apply the same climate sensitivity to all cases since we want to make the different literature sources comparable. As a result, we scale up the metric values in Sand et al. (2016) by about 34 percent, as the original climate sensitivity was 2.9 K.




Most emissions stay relatively constant throughout the year, while the changing seasons result in much larger emissions from the domestic sector in winter than in summer. We account for this seasonality for those metric datasets compatible with this, otherwise, annual emission and metric values are applied.

The global and regional temperature responses of Finnish emissions are estimated by combining AGTP and ARTP values with emissions. For an emission scenario E(t), the global temperature response is

$$\Delta T_{i,s,t}(t) = \int_0^t E_{i,s,t}(t') \times AGTP_{i,s,t}(t - t')dt' \qquad (5)$$

based on AGTP values. Similarly, the temperature responses in latitude bands can be estimated by replacing AGTP with
ARTP values. The global temperature response can also be estimated by using ARTPs and taking the area-weighted global mean. As the forcing-response coefficients are different and the ARTP concept can better parameterize varying efficacies, the estimate global temperature response may vary whether based on AGTPs or ARTPs.

Our climate impact dataset can be analyzed in many different dimensions, such as for different time scales, for different
emission sectors, for different processes, for pulse or scenario emissions. We show some examples. As we focus on near term climate change and the global and regional temperature, most of the discussion in this paper utilizes AGTP and ARTP for the mean warming in the first 25 years after a pulse emission, as recently proposed by Shindell et al. (2017). Mean(AGTP(1-25)) is the average temperature response over the time period, which differ from AGTP(25) being a snapshot at the time horizon of 25 years.

**3. Results**

**3.1 Emissions**

Fig.. 1 shows the Finnish emissions and their trends from 2000 until 2030 for the studied pollutants. Emissions by sector for 2000, 2010 and 2030 can be found in Table A1 of the Supporting material. Emission reductions are expected for practically all of the pollutants and greenhouse gases, especially between 2010 and 2030, but the magnitude differs between the species.
Reductions of CO2 and SO2 take place to large extent in the energy production sector following the reduction of energy consumption of fossil fuels, i.e. coal, oil and peat (Table 2).

CH4 emissions have declined mostly due to developments in waste sector. Amounts of methane recovered from landfills have increased during the study period following EU and national regulations. Methane emissions from landfills have also
declined because the energy use of municipal solid waste has increased instead of landfilling; a development that is expected



to continue also until 2030. Another factor explaining the declining emissions by 2030 is the prohibition of disposal of organic wastes to landfills after 2016.

The transport sector is responsible for the decline of the emissions of CO, NOX, VOC as well as the particle species, black carbon (BC) and organic carbon (OC). The modernization of the vehicle fleet and consequent introduction of stricter emission controls required by the EURO-standards explain the decline in CO, NOX and NMVOC emissions. The standards do not directly regulate BC or OC emissions, but since they are the main constituents of the regulated particulate emissions, reductions in emissions of BC and OC are expected, especially after the introduction of the diesel particulate filters for on-road light duty vehicles from 2010 onwards. The stoves and boilers in the residential sector will remain significant emitters 230 of several pollutants, since the regulation following the European Union Ecodesign directive will not have major impact by 2030, due to the relatively long lifetime of Finnish heaters (Savolahti et al. 2016).

NH3 and N2O emissions remain relatively stable throughout the study period, since either much of the emission reductions have already taken place before the study period or no major changes are expected in the main emission sectors (agriculture 235 for NH3).

**Figure 1. Finnish emissions (Gg a-1) of air pollutants and greenhouse gases in the period 2000 to 2030 in the baseline scenario. Emissions by sector for 2000, 2010 and 2030 can be found in Table A1 of the Supporting material.**

**3.2 Climate impact of Finnish emissions**

Figure 2 shows the 2010 emissions weighted with the global metrics, GWP and GTP, to CO2 equivalents using 10, 20, 50 and 100 year perspectives. Aamaas et al. (2013) studied global emissions with these metrics, while we focus in detail on Finnish emissions. In addition, we show the emission metric mean (GTP(1-25 yrs)), which gives the SLCFs a relatively large weight, similar to GTP(10 yrs) for the aerosols and in between GTP(10 yrs) and GTP(20 yrs) for CH4. In Figure 2 the 245 emissions are considered as a pulse and the figure does not take into account any emissions after 2010. Figure 2 demonstrates that the SLCFs have a larger relative importance with the metrics for shorter time horizons. However, in all cases CO2 still is the most important species. With the emission metric with the 10 year horizon (GTP10) the warming SLCFs comprise more than two thirds of the warming effect of CO2, but overall the net-impact of all short-lived species is about 30 percent of CO2, due to the partly counteracting cooling effect of NH3, SO2, NOX and OC. The relative importance 250 of the SLCFs decreases with time, especially with GTP, as expected, and the relative effect is lowest with the temperature change metric with 100 year time horizon (GTP100), being about 6 percent of CO2. Among the non-CO2 emissions, the relative impact of N2O increases with increasing time horizon due to the much longer atmospheric lifetime than for the other pollutants.



**Figure 2. Finnish 2010 emission (Mt CO2-eq) as a pulse emission weighted by various global metrics. CO2 is separated out and the net impact of the non-CO2 is given by the star.**

As we focus on near term climate change and the global and regional temperature, the remaining paper utilizes AGTP and ARTP with a time horizon of mean(GTP(1-25 yrs)), as proposed by Shindell et al. (2017). The rest of the paper is mostly applying ARTP values, following the argumentation by Aamaas et al. (2017) that ARTPs may give a better estimate of the global impact than AGTPs since they account for varying efficacies with latitude to a larger degree. The AGTP(1-25 yrs) and ARTP(1-25 yrs) used in this study are presented in Tables A2 and A3 of the Supporting material.

### 3.2.1 Climate impacts by emission sector

This section discusses the global temperature response of the emissions by pollutant and emission sector based on weighted ARTP values (Aamaas et al., 2017). The general findings described in the following paragraphs would be similar with AGTPs, and similar figures based on AGTP values (Aamaas et al., 2016) are given in Figure A1 of the Supporting material for comparison. Figure 3A, 3B, and 3C show the warming due to emissions in 2000, in 2010, and in 2030 following the baseline projection, respectively. The sum of all sectors is given in Figure 3D. The pollutant mix varies for the different sectors. CO2 is the most important pollutant for combustion in energy production and industry (ENE IND) and road transport (TRA RD), while methane is most important for waste (WST) and agriculture (AGR) sectors. BC emissions cause more than half of the warming (about two thirds in 2030) in the domestic sector (DOM) and a significant share of the warming in the on-road (TRA RD) transport as well as off-road transport and machinery (TRA OT) sources. The importance of BC is somewhat smaller if we apply AGTPs. Rest of the warming effect for these sectors is due to CO2 emissions from fossil fuels, especially diesel and light fuel oil. Wood is an important fuel in the domestic sector, and since this study considers wood fuel as CO2 neutral, the CO2 warming effect is not as pronounced as, for example, in the on-road transport sector. Organic carbon has been the most important cooling agent in domestic and the transport sectors, as fuelwood does not contain much sulfur, and it has been phased out from liquid fuels in the transport sector as well. Overall, SO2 is the major cooling pollutant, mainly due to emissions from energy production (ENE IND) and industrial processess (PROC). Agriculture is an important source of ammonia (NH3), which has a cooling effect (Fig. 4) via its participation in formation of cooling atmospheric aerosols like ammonium sulphates and nitrates.

Year 2000 was relatively warm and 2010 relatively cold in Finland, which is reflected as a higher use of coal, peat and wood fuels in 2010, and consequently also as higher emissions for some species. From 2000 to 2010, CO2 emissions from ENE IND increased by 22 percent and BC emissions from DOM by 37 percent. However, because of additional mitigation measures following legislation, CH4 emissions from the WST decreased by 38 percent. Also, despite the higher fuel use,



improved flue gas cleaning measures caused SO2 emissions in ENE IND to decrease by 18 percent. On the other hand, the reduction of SO2 has increased the warming effect of the ENE IND sector in 2010 compared with 2000. The increasing SLCF emissions in the DOM sector, particularly black carbon, have led to additional net warming despite the fact that the
organic carbon emissions offset about a fifth of the black carbon effect in both years. The decreasing trend for the use of heating oil in the domestic sector has reduced CO2 emissions between 2000 and 2010. Emissions from the PROC sector are relatively neutral in terms of their climate effect. In general, taking into account all sectors, the emission changes between 2000 and 2010 in Finland have led to net-warming (increase by 3 percent), mostly due to the increase of CO2 emissions (warming) and decrease in SO2 emissions (warming) from the ENE IND sector, which have offset the reduction of CH4
emissions (cooling) in the WST sector.

The baseline projection will lead to emission reduction of all pollutants between 2010 and 2030, from more than a 50 percent reduction of BC to a small reduction for N2O (Fig. 1 and Table A1). Because of climate policies, CO2 is reduced following the declining use of fossil fuels (Table 2, Fig. 3B, 3C and 3D). The SO2 emissions continue their decline between 2010 and
2030, particularly in the ENE IND sector, which leads to additional warming, but only partly offsetting the reduced CO2 (Fig. 3B, 3C and 3D). In the on-road (TRA RD) and off-road (TRA OT) transport sector, particularly the warming effect from the SLCFs declines, because the new vehicles, in order to comply with the European emission legislation, are equipped with efficient emission reduction technologies (Fig. 3B and 3C). The amount of domestic wood combustion is expected to decrease in the baseline due to improved energy efficiency in housing, which is the main reason for the reduced SLCF
emissions in the sector (Fig. 3B and 3C). However, when interpreting these results it is important to note that the prevalence of domestic wood combustion has been increasing during the 2000's and the future wood use in households is challenging to predict. Therefore the emissions from the domestic sector should be considered uncertain. This is demonstrated in a sensitivity analysis of future particle emissions from the domestic sector presented by Savolahti et al. (2016). Also the methane emissions in the WST sector continue their decline (Fig. 3B and 3C). As a consequence of the emission changes,
the net-temperature impact of 2030 emissions is 32 percent lower compared with the 2010 emissions (Fig. 3D). Practically all sectors but AGR contribute to the reduced warming (Fig. 3B, 3C).

**Figure 3. The temperature response (µK) due to emissions in 2000 (A), 2010 (B), and 2030 (C) from sectors energy and industry (ENE IND), industrial processes (PROC), transport road (TRA RD), off-road transport and machinery (TRA OT), domestic**
**(DOM), waste (WST), agriculture (AGR), and other (OTHER). The sum of all sectors is shown in (D). The climate metric applied is the global mean(ARTP(1-25 yrs) for pulse emissions.**

### 3.2.2 Cumulative temperature development 2000-2030

So far we have shown the global temperature response for pulse emissions, while the temperature impact over time will
depend on continuous emissions from every single year. The cumulative global temperature impact by pollutants and sectors



for Finnish emission in 2000-2030 is shown in Fig. 4, based on ARTPs in Aamaas et al. (2017). Similar figures based on AGTP values (Aamaas et al., 2016) are given in Figure A2 of the Supporting material. Fig. 4 demonstrates why emission reductions of CO2 and other long-lived greenhouse gases are key for limiting the long-term surface temperature increase. As more years are added, the relative importance of CO2 increases, since a large portion of it stays in the atmosphere for

hundreds of years. This relative importance over time also occurs in case of N2O. The air pollutants become of less relative significance with time, which is mostly because of those pollutants being quickly removed from the atmosphere, but also because of the reduced emissions levels in the later period. Almost all sectors have a net-warming temperature response, with the exception of cooling from ENE IND sector in the first ten years and a slight cooling from PROC sector nearly until 2030 (Fig. 4B). Cooling from mainly SO2 emissions is offsetting the warming impact of CO2 from those sectors. Over time,

ENE IND becomes the most influential sector, being the single largest contributor of CO2. BC is the most significant warming pollutant in the domestic sector and CH4 for the agriculture and waste sectors.

**Figure 4: The global temperature development (mK) of Finnish emissions for the period 2000-2030. Temperature is given by pollutants in (A) and by sectors in (B).**


### 3.2.4 Estimated climate impacts depend on the chosen metrics

The estimated temperature response depends on the metric parameterization applied. In Fig. 5, we compare how the results applying Finnish SLCF emissions for the year 2010 during summer (May-October) and winter (November-April) vary for two different metric approaches. We compare global temperature responses using AGTPs from Aamaas et al. (2016) and

ARTPs from Aamaas et al. (2017). A decomposition into different processes is also included. Some pollutants have both warming and cooling processes, such as BC and NOX. The same process can be warming for one pollutant and cooling for another, for example NOX emissions remove CH4 from the atmosphere (cooling), while VOC and CO emissions add CH4 (warming). Changes in the methane concentration will also influence ozone, giving rise to the methane-induced ozone effect and reinforcing the methane effect.


Since the ARTPs account for varying efficacies with latitude, Aamaas et al. (2017) argue that ARTPs may give a better estimate of the global impact. However, as the regional response coefficients behind the ARTP studies are mostly built on results from one model, we acknowledge that our results have potentially significant uncertainties. The differences between the temperature impact of summer and winter emissions in Fig. 5 are not only caused by different climate metric values, but

also that the domestic emissions are largest in winter (>70 percent of annual emissions). Applying AGTPs or ARTPs give mostly similar results, with the exception of BC, especially for emissions occurring in winter. For a ton of BC emission with the AGTP metric the summertime warming impact is 16 percent higher compared with winter whereas with ARTP the wintertime impact is higher by more than 90 percent. The difference is driven by a more detailed parameterization of the



effect of BC deposition in the Arctic. In case of ARTP more than 70 percent of the net impact for winter emission comes for

BC deposition on snow. The annual impact of winter emissions of BC is 59 percent with the AGTPs but 76 percent with the

ARTPs. From a mitigation perspective, stronger attention should be placed on reducing winter emissions of BC when

applying ARTPs rather than the AGTPs.

**Figure 5: The global temperature response (µK) of Finnish emissions in 2010 by applying the mean temperature 1-25 yrs after the pulse emission. This figure compares emissions occurring in summer vs. winter, as well as global temperature estimated by either AGTP (Aamaas et al., 2016) or ARTP (Aamaas et al., 2017). The emission region for the climate metrics is Europe for all cases. We present four cases for each pollutant from left to right: 1) summer emissions with AGTP, 2) winter emissions with AGTP, 3) summer emissions with ARTP, 4) winter emissions with ARTP. The responses are divided into six different processes.**

**3.2.5 Arctic temperature response from Finnish emissions**

We continue the comparisons of different climate metric studies, focusing on SLCFs, with the attention on the Arctic area. Finland is closely situated to the Arctic as practically the whole country is north of the 60°N latitude and a significant area lies north of the Arctic Circle. Fig. 6 shows the Arctic (between 60° to 90°N) temperature response based on the ARTP metrics from Aamaas et al. (2017). As a general observation the temperature responses are larger in the Arctic (Figure 6)

compared with the global ones (Figure 5). Emissions of BC become even more important in the Arctic perspective, especially for Finnish emissions during winter (82 percent of the annual impact). More than 80 percent of the net impact for winter emission comes for BC deposition of snow. For emissions in summer, the cooling from SO2 is larger than the warming from BC.

**Figure 6: The temperature response (µK) in the Arctic of Finnish emissions in 2010, by applying the mean temperature 1-25 yrs after the pulse emission. This figure compares emissions occurring in summer (S) vs. winter (W) by applying ARTP values (Aamaas et al., 2017). The emission region for the climate metrics is Europe for all cases.**

As the Sand et al. (2016) study also provides ARTP values for BC, OC and SO2 in the Arctic (Table A3 of the

Supplementary material), we compare the annual temperature responses in the Arctic for those pollutants with input from Aamaas et al. (2017). Unfortunately, Sand et al. (2015) did not include climate metric values for other pollutants. Three differences between the two studies were identified, some of which led to adjustments in the parametrizations, namely: (1) Aamaas et al. (2017) provided ARTPs for emissions in Europe whereas Sand et al. (2016) analyzed the temperature impacts of emissions from the Nordic countries; (2) The climate sensitivities in the studies are different, and we adjusted the climate

sensitivity upwards in the Sand et al. (2016) study to make its parameterization more comparable to the ones in Aamaas et al. (2017) and used in this study; Finally (3) the aggregation of atmospheric forcing processes between Aamaas et al. (2017) and Sand et al. (2016) is different, such as Sand et al. (2016) do not account for the semi-direct effect from BC. In Fig. 7 we have summed the direct and indirect effects in Sand et al. (2016) into "aerosol effects";





Fig. 7 shows that BC emissions from Finland lead to significant warming with both parameterizations. However, the net-impact of BC emissions with the adjusted Sand et al. (2016) approach is 18 percent higher compared with Aamaas et al. (2017). The largest difference is for SO2, where Sand et al. (2016) estimate a much larger indirect effect (Fig. 7). Unfortunately, Sand et al. (2016) did not provide global temperature responses. But we can see that in short term, these larger sensitivities for SO2 would probably lead to a negative temperature response for the PROC sector (see Fig. 3), while

the warming of CO2 would to a larger extent be counteracted by cooling from SO2 in the ENE IND sector. As the magnitude of this indirect effect is uncertain, the difference in the two estimates gives an indication of the uncertainty. The Arctic temperature response is also larger for OC (by 58 percent) than estimated based on Aamaas et al. (2017).

As this study focuses on Finland, the ARTPs in Sand et al. (2016) could be considered more representative. However,

differences between the two studies and as a consequence in the results may also be due to different study designs, such as partly different selection of climate models, and not necessary only a result of a different representation of the geographical location of the source region.

**Figure 7: The temperature response (µK) in the Arctic (60-90° N) due to Finnish emissions of BC, OC, and SO2. The column to the left for each pollutant is based on ARTPs for Europe by Aamaas et al. (2017) and to the right ARTPs for Nordic countries by Sand et al. (2016).**

Fig. 8 compares global and Arctic temperature responses to Finnish emissions, using the Aamaas et al. (2017) approach. It demonstrates that the temperature response in the Arctic is typically stronger than the global average. If we apply the ARTP

methodology for GHGs, the response in the Arctic is up to 50% larger than the global average due to stronger local feedback processes in the Arctic (Boer and Yu, 2003). The ozone precursors have similar or weaker efficacies in the Arctic compared with the GHGs. However, the aerosols and sulfur emissions stand out (Fig. 6 and 7). Applying this method, Finnish emissions of BC, SO2, and OC have a 310%, 120%, and 100% stronger efficacy in the Arctic than the global average (Fig. 6 and 7). For BC, this amplification in the Arctic is even stronger for emissions occurring in winter. Hence, the results indicate

that mitigation of Finnish BC emissions is especially beneficial for limiting Arctic warming.

**Fig 8. Global and Arctic (60-90° N) temperature responses (µK) to Finnish emissions based on ARTP values in Aamaas et al. (2017). As for most of the figures, the temperature response is the mean response 1 to 25 years after a pulse emission.**

### 4. Discussion and Conclusions

The first objective in our study was to produce an integrated multi-pollutants emission dataset for Finland for 2000 to 2030. We were able to achieve this aim, but it required the use of several data sources and studies that are not necessarily



maintained at a regular basis. Future efforts should pursue to maintain the integrated multi-pollutant database developed for this work. This would require an integrated modelling environment, for example the Finnish Regional Emission Scenario model (FRES), and further work to fill in the gaps for the missing sectors and pollutants via developing relevant activity and
emission factor databases into the FRES framework.

Our second set of objectives for this study was to compare different climate metrics and to assess their suitability for calculating the climate impact of a multi-pollutant emission set. Several air pollutants and greenhouse gases have detrimental impacts on global and regional climate, human health and wellbeing as well as crop yields (see i.e. Shindell et al. 2012).
Since the magnitudes and pathways of the effects differ between the constituents, integrated modelling is needed to understand the consequences and form the basis for robust climate and air quality policies. This paper applied and compared various climate metrics to study the approximate integrated climate impact of Finnish air pollutant and greenhouse gas emissions globally and in the Arctic area. The results demonstrated that the relative impacts and importance of individual species as well as sectors can differ significantly between the studied temporal response scales, emission seasons as well as
geographical response scales. Especially the warming or cooling impact of SLCFs is sensitive to the studied time scale, with shorter time spans showing greater importance compared with GHGs.

Finnish emissions and their climate responses are relatively small; therefore it is challenging to use climate models to study the climate effect of national policies and to analyze the role of each pollutant and sector. This study demonstrated a method
to overcome this challenge by utilizing emission metrics. All studied metrics provided interesting insights into the impacts of Finnish emissions and which aspects could be emphasized when formulating mitigation strategies. We assessed that particularly the AGTP and ARTP based metrics provided useful information, although one should not rule out the significance of the other radiative forcing based metrics due to their relevance in connection with climate change mitigation work of the UNFCCC and IPCC. We preferred to use ARTP approaches to assess the impacts of Finnish emissions to both
global and Arctic climate, because it includes the regional or latitudinal dimension of emission impacts in more detail. We also chose to use the mean(1-25 yrs) timeframe, since for the time being there is no established climate metrics for air pollutants, and this approach was recently suggested by Shindell et al. (2017) to be used in connection with SLCFs. To our knowledge, we are the first to present metric values for this climate metric.

The use of ARTPs to study the impacts of Finnish emissions is useful for designing national emission mitigation strategies also from a regional perspective. Finland is an Arctic country and a member of the Arctic Council, which is why there is high interest on understanding the Arctic impacts.

The third set of objectives was to estimate the climate impact of Finnish air pollutants and greenhouse gases utilizing the
selected metrics. Our analysis across climate metrics, time horizons, pollutants and Finnish emission pathways demonstrated



that carbon dioxide emissions have the largest climate response also in the short term, and its relative importance increases the longer the time span gets. Hence, mitigation of carbon dioxide is crucial for reducing the climate impact of Finnish emissions. In the near or medium term, i.e. 25 year perspective, especially methane and black carbon have relatively significant warming impacts additional to those of carbon dioxide. SO2 on the other hand is an important precursor to light
reflecting sulphate aerosol thus having a cooling impact, offsetting part of the warming impact of the other species.

Of Finnish emissions, the combustion in energy production and industry has the largest global temperature impact in the medium and long term due to biggest carbon dioxide emissions, while sulfur dioxide emissions induce a shorter term cooling. Transport has the second biggest warming impact, and although that is expected to decrease notably by 2030 due to
stricter control on particulate and consequently black carbon emissions, it will remain a major source of carbon dioxide. Emissions from domestic and agriculture sectors also have a considerable warming impact, and they will remain so, due to the relatively large respective emissions of black carbon and methane from the combustion of solid fuels, especially wood.

For all of the species the temperature response of Finnish emissions is generally stronger in the Arctic than globally, but
most significantly so in case of black carbon and sulfur dioxide. Results obtained with the ARTP metric indicated that especially mitigation of wintertime black carbon emissions are important for reducing the temperature increase in the Arctic. Emissions of sulfur dioxide are expected to continue decreasing and this has many benefits (Ekholm et al. 2014). However, it will offset some of the climate benefits of the reduced carbon dioxide emissions, and this should be taken into consideration in climate assessments.

The fourth major objective of this study was to recommend a set of global and regional climate metrics to be used in connection with Finnish SLCF emissions. In this paper we provide a comparison and discussion of several climate metrics to be used in connection with Finnish SLCF emissions mostly relying on those presented in Aamaas et al. (2017) that in our understanding is currently the most complete set of climate metrics available for assessing the global and Arctic temperature
responses of European emissions. For the GHGs, we argue to apply the metric parameterization from IPCC AR5 (Myhre et al., 2013), but with an upward revision for CH4 (Etminan et al., 2016). The coefficients for mean(ARTP(1-25 yrs)) (see also Shindell et al. 2017) in Table 3 have been evaluated to be useful for assessing different mitigation pathways in a 25 year time span. This time window is relevant for policies that focus on reducing global or Arctic warming by 2040 or 2050. Corresponding mean(RTP(1-25 yrs)) values are shown in Table A2.

Table 3. The climate metric values (°C/Tg) used in this study. Mean(ARTP(1-25yrs)) and mean(AGTP(1-25yrs)) values for SLCF and GHG emissions. The Arctic response for the GHGs is based on the latitudinal pattern for CH4. The annual average is based on emissions in 2010. Normalized values (CO2-equivalents) are shown in Table A2.



The assessed temperature impact of an emission dataset depends on the set of metrics available, as well as the applied metric setup, which bring uncertainties to the results. As there is no consensus on one individual set of metrics, especially in case of air pollutants, the results will differ between different studies. This work estimated the global and regional temperature impacts of Finnish emissions based on methodologies in three recent papers. As all of these studies utilize partly the same radiative forcing datasets and partly similar general circulation models and chemistry transport models, the uncertainties may

be in fact be larger than our results indicate if a larger set of background studies would be utilized. Future work should continue to explore these uncertainties and provide improved metrics.

Since the atmospheric lifetime of SLCFs is relatively short, their climate impact is more dependent on the emission region than with GHGs. Using Europe as a proxy for the emission region, as in this study, gives us a more representative picture of

the Finnish case than would the global average. Further development of the metrics should use more precisely the geographical location of Finland as the emission region in order to provide more precise temperature estimates for the Finnish emissions. This is mostly because the snow albedo effect of BC emissions is expected to be larger for Finland, compared with to the source regions used in our study. This is indicated by a study for Norway by Hodnebrog et al. (2014). Future work should also focus on providing metrics for potentially missing species that could be important, for example dust

aerosol.

Scientific literature has demonstrated that the climate impact of biomass combustion may depend on the timescale and forestry practices (i.e. Cherubini et al. 2011, Repo et al. 2012 and Repo et al. 2015), which have not been a focus of this study. Since the use of biomass for energy is important in Finland and will likely remain so in the coming decades, future

studies could utilize metrics to study its climate impacts. This study has mostly focused on surface temperature metrics, however interesting other impacts could be studied using the metric approach. For example Shine et al. (2015) has recently presented a new metric named the Global Precipitation change Potential (GPP), which is designed to gauge the effect of emissions on the global water cycle.

The improved understanding of the impact pathways of different pollutants has improved in recent years, which further has led to revisions of the climate impact estimates. Such development is expected to continue. The metric studies, however, are often based on earlier RF studies, and a time lag from new scientific understanding to this being reflected in the climate metrics exists. This study has utilized the latest metric studies, but there are already studies available, for instance on BC, indicating that the temperature response may be smaller than in this work (e.g., Stjern et al., 2017). As the understanding of

the climate system improves, the estimated we give here for Finland should be updated.




## Author contribution

KJK and MS compiled the emission data with supporting contributions from NK and VP. BA prepared the climate metrics databases and applied them to the emission data. KJK, BA and MS were leading the preparation of the manuscript. NK and VP acted as contributing authors.

## Acknowledgements

This study has been financially supported by the Finnish Ministry of the Environment and the Ministry for Foreign Affairs of Finland via the Baltic Sea, Barents and Arctic region cooperation programme (IBA, decision HEL8118-34), by the Academy of Finland project grants NABCEA (Novel Assessment of Black Carbon in the Eurasian Arctic: From Historical Concentrations and Sources to Future Climate Impacts; decision no 296644), WHITE (Keeping the Arctic White: Regulatory Options for Reducing Short-Lived Climate Forcers in the Arctic; decision no 286699) and BATMAN (Environmental impact assessment of airborne particulate matter: the effects of abatement and management strategies; decision no 285672) as well as by NordForsk under the Nordic Programme on Health and Welfare, project grant NordicWelfAir (Understanding the link between air pollution and distribution of related health impacts and welfare in the Nordic countries; #75007). Borgar Aamaas has been funded by the European Union Seventh Framework Programme (FP7/2007-2013) under grant agreement no 282688 – ECLIPSE.

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

**Tables & figures**

Table 1. Data sources of the historical emission data for 2000-2010

| Pollutant | Data source |
|---|---|
| Black carbon (BC), organic carbon (OC) | FRES model |
| CO | GAINS model (http://gains.iiasa.ac.at) |
| $CO_2$, $CH_4$ and $N_2O$ from combustion sources | FRES model |
| $CO_2$, $CH_4$ and $N_2O$ from other sources than combustion | National inventory of greenhouse gases specified in the Kyoto Protocol to the Secretariat of the UNFCCC |
| $NH_3$ and VOC | National emission inventory to the UNECE Convention on Long-Range Transboundary Air Pollution (CLRTAP) |

Table 2. Primary energy consumption in Finland (TWh $a^{-1}$) (Ministry of Employment and the Economy, 2013)

|  | 2010 | 2020 Baseline | 2030 Baseline |
|---|---|---|---|
| Traffic fuels | 50 | 48 | 42 |
| Other oil fuels | 48 | 43 | 32 |
| Coal | 52 | 50 | 22 |
| Gas | 41 | 37 | 31 |
| Peat | 26 | 16 | 13 |
| Wood fuels, | 89 | 98 | 101 |
| -of which RWC | 19 | 15 | 17 |
| Nuclear power | 66 | 106 | 171 |
| Hydro power | 13 | 14 | 15 |
| Wind power | 0.3 | 6 | 7 |
| Others, including waste | 10 | 16 | 19 |





| Import of electricity | 11 | 0 | -3 |
| Sum | 407 | 433 | 459 |

Table 3. The climate metric values (°C/Tg) used in this study. Mean(ARTP(1-25yrs)) and mean(AGTP(1-25yrs)) values for SLCF and GHG emissions. The Arctic response for the GHGs is based on the latitudinal pattern for $CH_4$. The annual average is based on emissions in 2010. Normalized values ($CO_2$-equivalents) are shown in Table A2.

| | Mean(1-25yrs), global response in °C/Tg | | | Mean(1-25yrs), Arctic response in °C/Tg | | |
|---|---|---|---|---|---|---|
| | Annual average | Summer | Winter | Annual average | Summer | Winter |
| $CO_2$ [$CO_2$] | 5.7E-7 | 5.7E-7 | 5.7E-7 | 8.2E-7 | 8.2E-7 | 8.2E-7 |
| $CH_4$ [$CH_4$] | 4.8E-5 | 4.8E-5 | 4.8E-5 | 6.9E-5 | 6.9E-5 | 6.9E-5 |
| $N_2O$ [$N_2O$] | 1.5E-4 | 1.5E-4 | 1.5E-4 | 2.1E-4 | 2.1E-4 | 2.1E-4 |
| $NO_X$ [$NO_2$] | -1.7E-5 | -2.3E-5 | -1.1E-5 | -1.9E-5 | -2.7E-5 | -1.1E-5 |
| VOC [VOC] | 9.6E-6 | 1.4E-5 | 6.1E-6 | 1.6E-5 | 1.6E-5 | 1.6E-5 |
| CO [CO] | 4.1E-6 | 3.9E-6 | 4.3E-6 | 5.2E-6 | 5.0E-6 | 5.4E-6 |
| BC [C] | 1.8E-3 | 1.1E-3 | 2.2E-3 | 7.4E-3 | 3.5E-3 | 9.8E-3 |
| OC [C] | -4.0E-4 | -5.6E-4 | -3.0E-4 | -8.1E-4 | -1.2E-3 | -5.8E-4 |
| $SO_2$ [$SO_2$] | -2.0E-4 | -3.1E-4 | -9.1E-5 | -4.4E-4 | -7.0E-4 | -1.9E-4 |
| $NH_3$ [$NH_3$] | -3.7E-5 | -4.5E-5 | -2.9E-5 | -6.2E-5 | -7.5E-5 | -4.8E-5 |






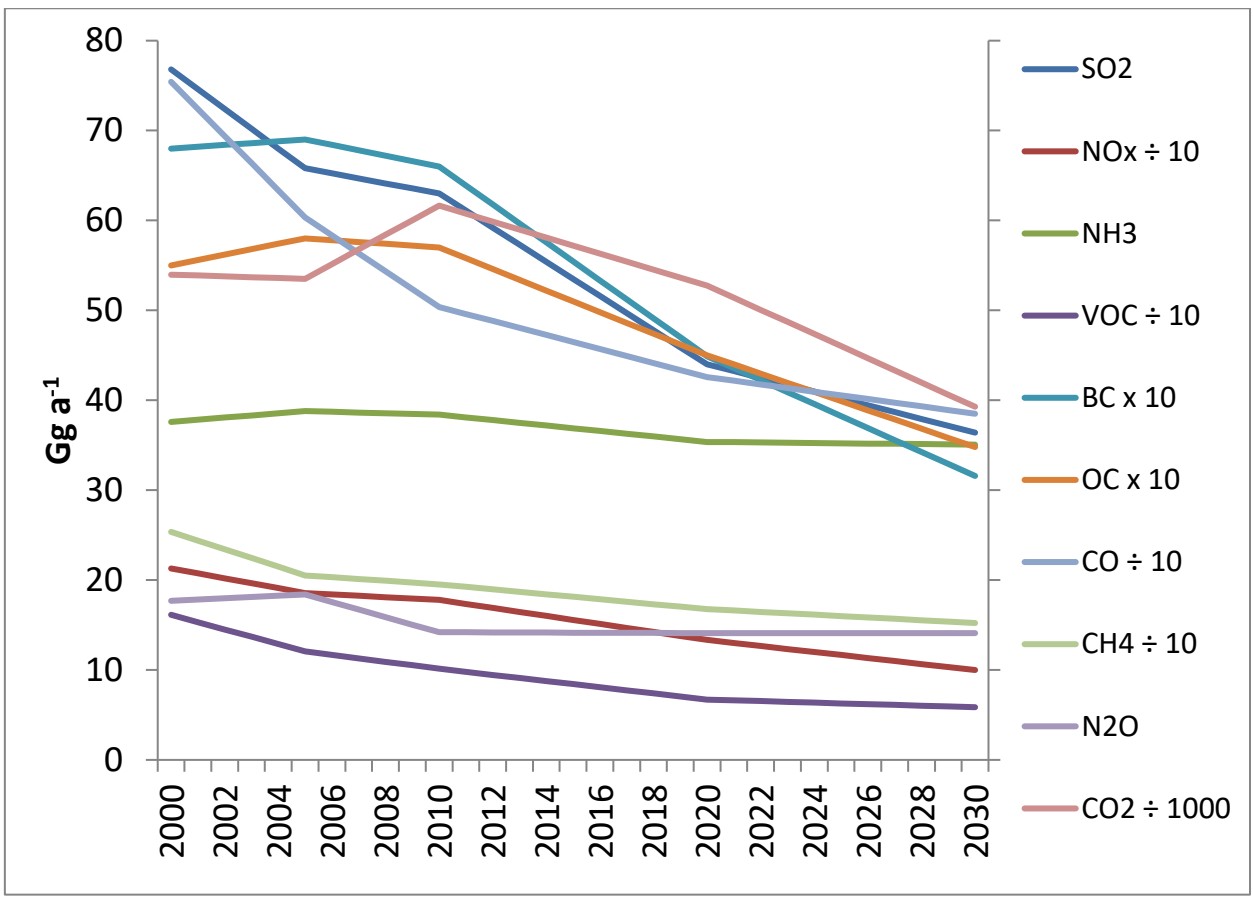

**Figure 1. Finnish emissions (Gg a⁻¹) of air pollutants and greenhouse gases in the period 2000 to 2030 in the baseline scenario. Emissions by sector for 2000, 2010 and 2030 can be found in Table A1 of the Supporting material.**






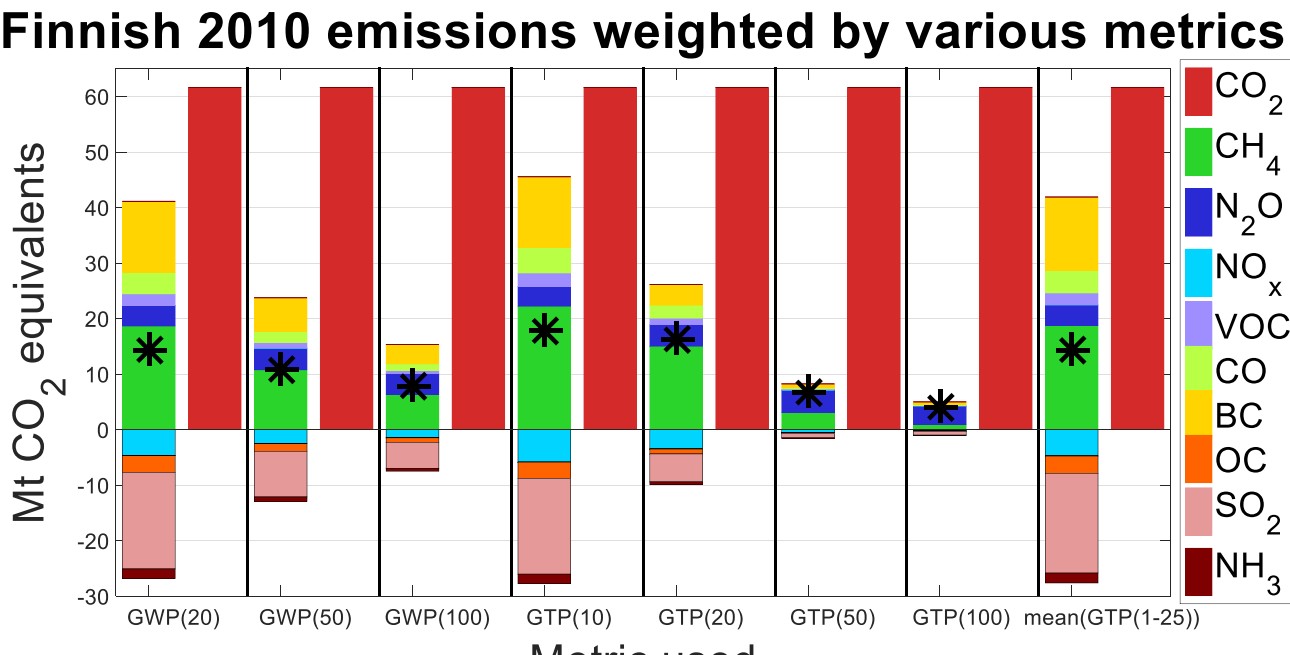

**Figure 2.** Finnish 2010 emission (Mt CO₂-eq) as a pulse emission weighted by various global metrics. CO₂ is separated out and the net impact of the non-CO₂ is given by the star.

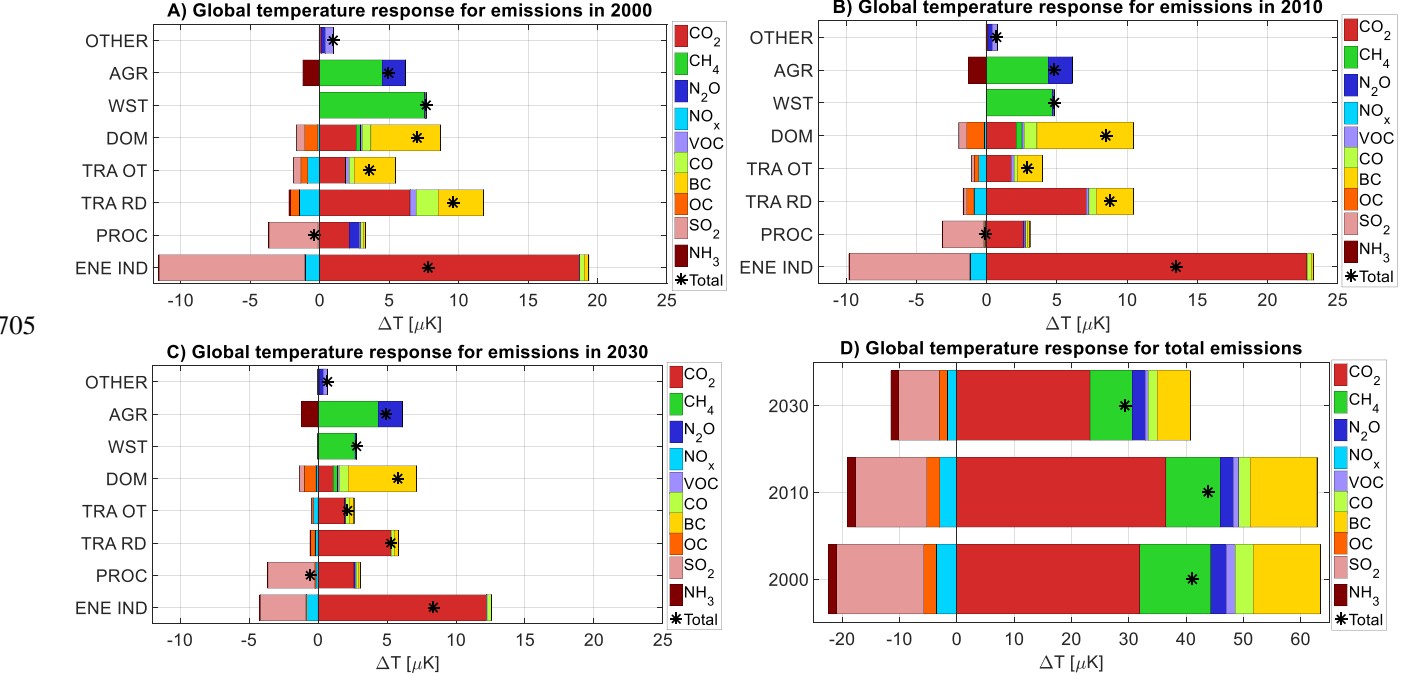





**Figure 3. The temperature response (µK) due to emissions in 2000 (A), 2010 (B), and 2030 (C) from sectors energy and industry (ENE IND), industrial processes (PROC), transport road (TRA RD), off-road transport and machinery (TRA OT), domestic (DOM), waste (WST), agriculture (AGR), and other (OTHER). The sum of all sectors is shown in (D). The climate metric applied is the global mean(ARTP(1-25 yrs) for pulse emissions.**

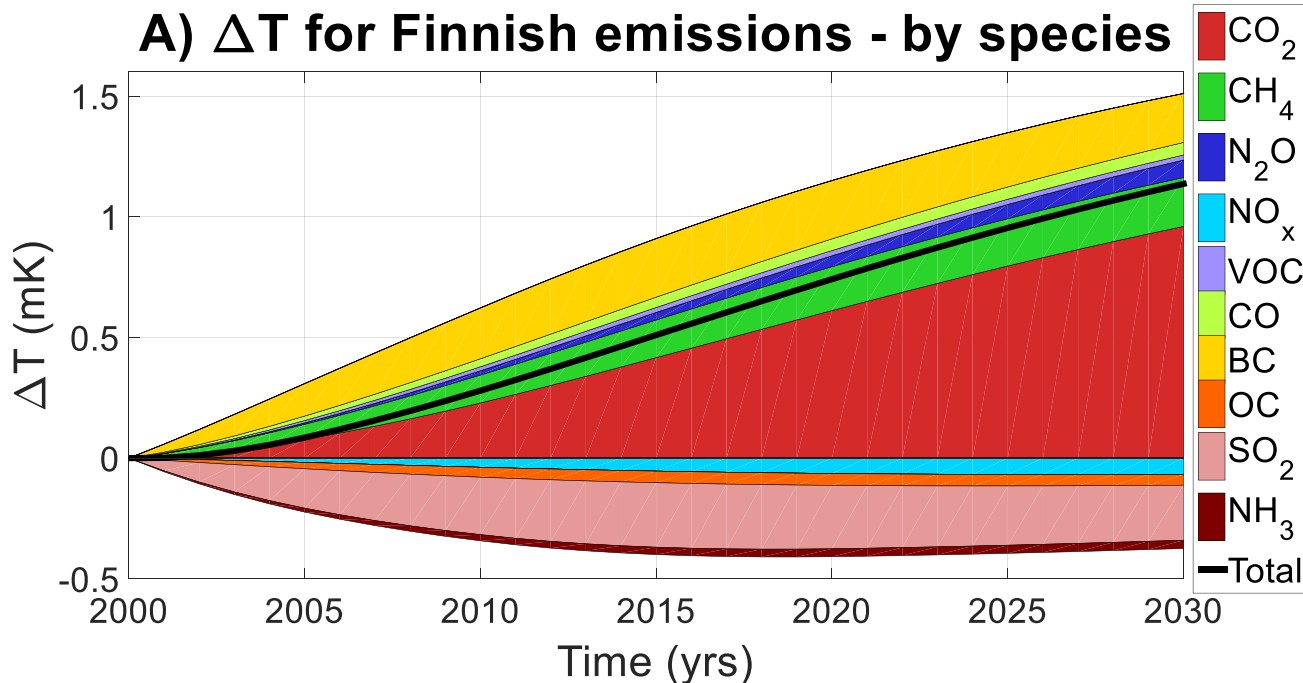



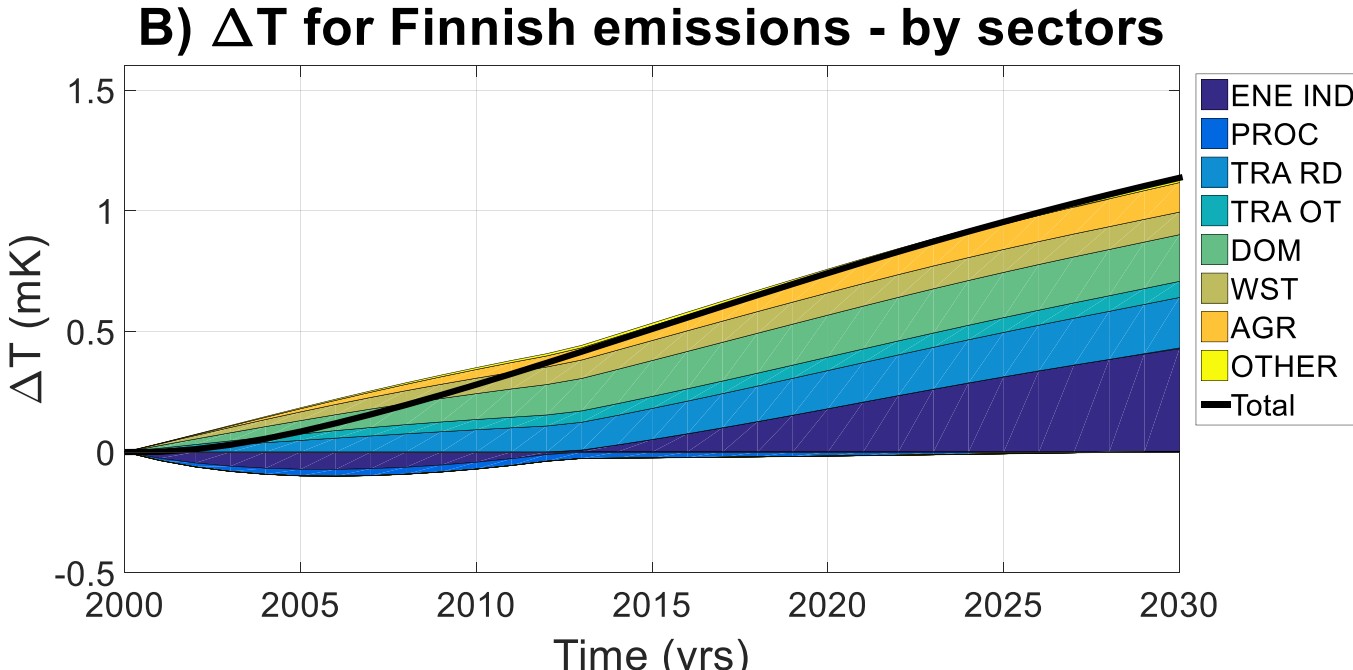

**Figure 4: The global temperature development (mK) of Finnish emissions for the period 2000-2030. Temperature is given by pollutants in (A) and by sectors in (B).**


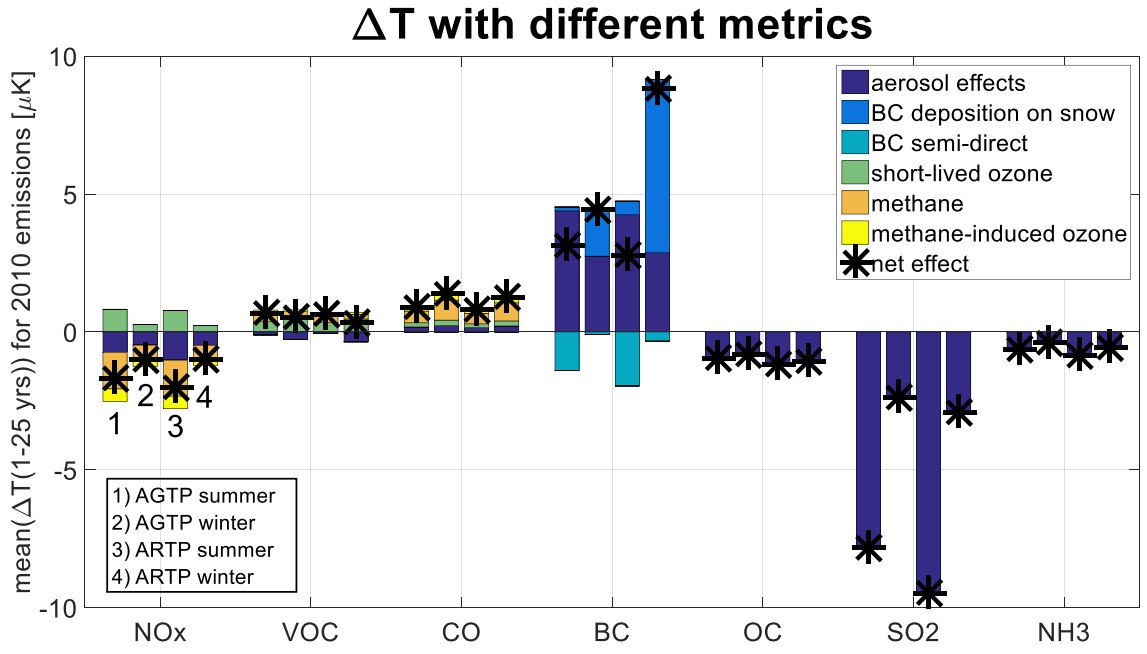





**Figure 5: The global temperature response (μK) of Finnish emissions in 2010 by applying the mean temperature 1-25 yrs after the pulse emission. This figure compares emissions occurring in summer vs. winter, as well as global temperature estimated by either**
**AGTP (Aamaas et al., 2016) or ARTP (Aamaas et al., 2017). The emission region for the climate metrics is Europe for all cases. We present four cases for each pollutant from left to right: 1) summer emissions with AGTP, 2) winter emissions with AGTP, 3) summer emissions with ARTP, 4) winter emissions with ARTP. The responses are divided into six different processes.**

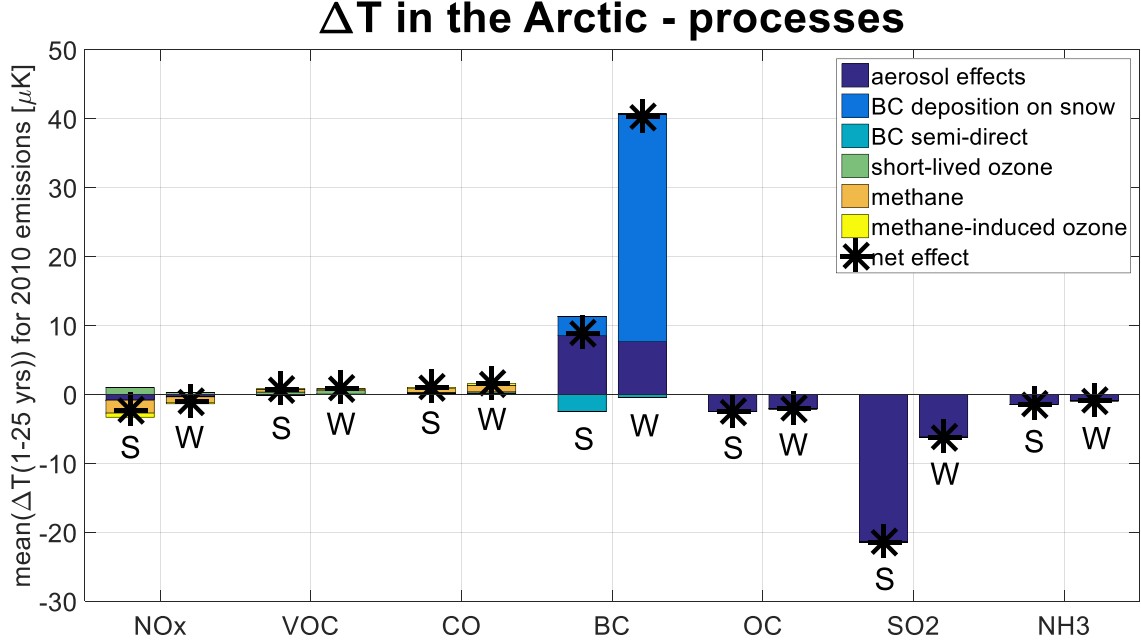

**Figure 6: The temperature response (μK) in the Arctic of Finnish emissions in 2010, by applying the mean temperature 1-25 yrs after the pulse emission. This figure compares emissions occurring in summer (S) vs. winter (W) by applying ARTP values (Aamaas et al., 2017). The emission region for the climate metrics is Europe for all cases.**



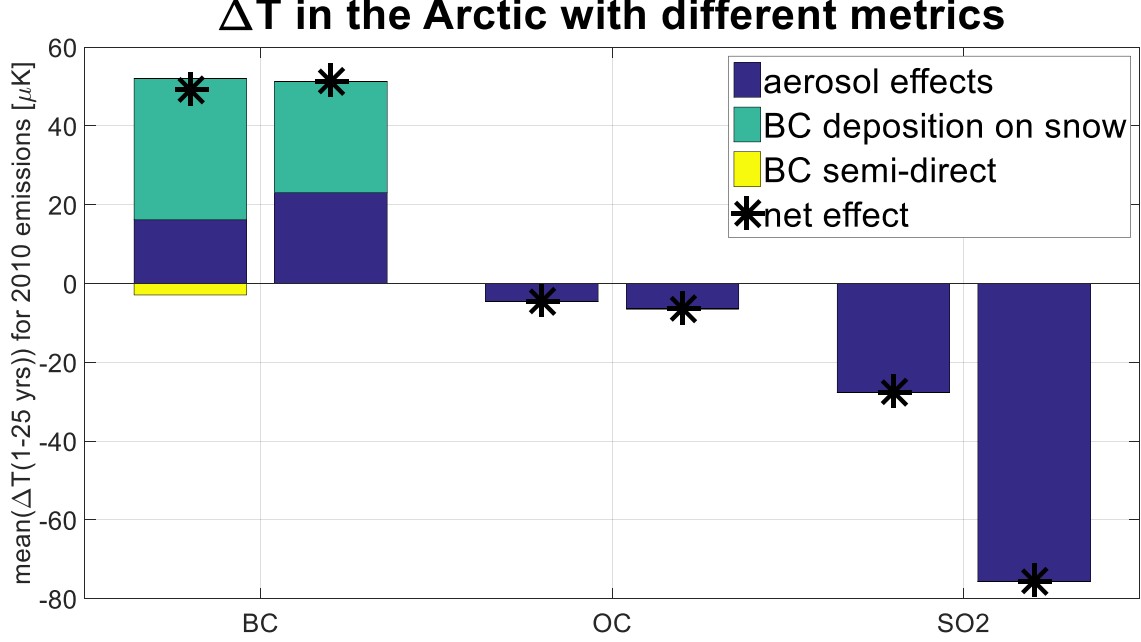

**Figure 7: The temperature response (µK) in the Arctic (60-90° N) due to Finnish emissions of BC, OC, and SO₂. The column to the left for each pollutant is based on ARTPs for Europe by Aamaas et al. (2017) and to the right ARTPs for Nordic countries by Sand et al. (2016).**





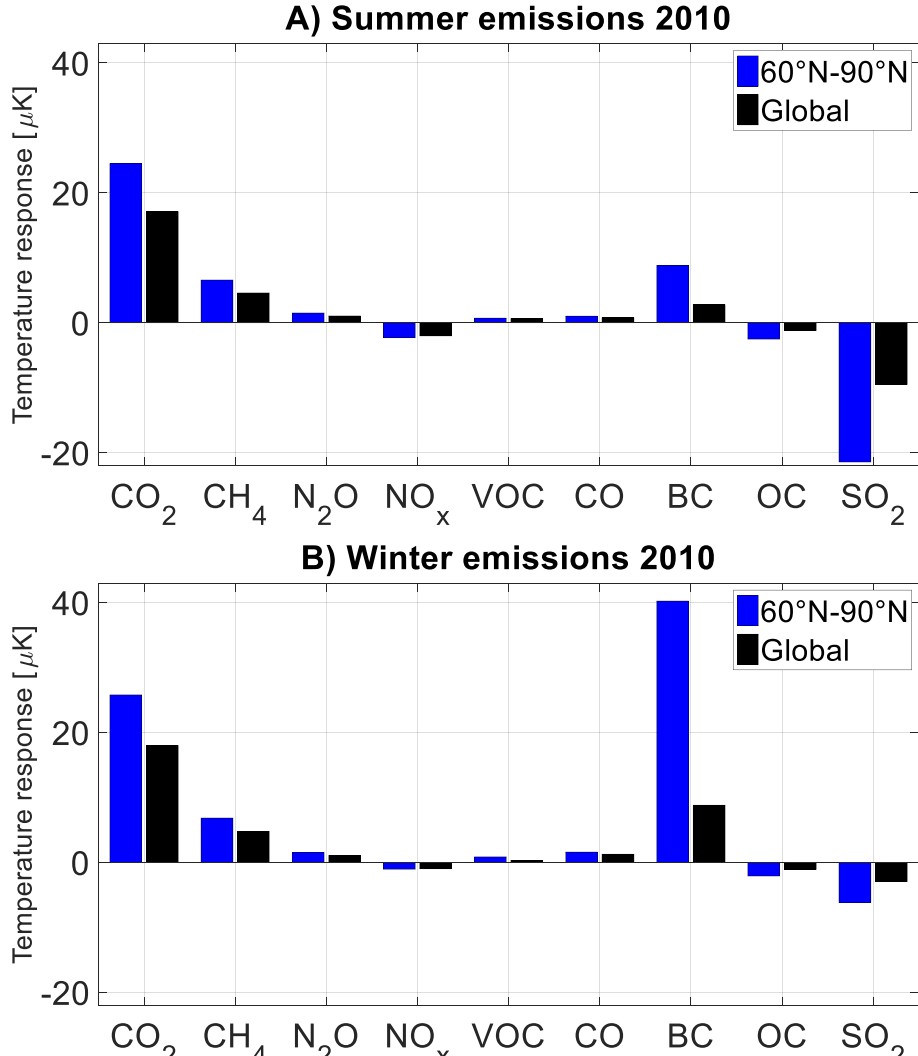






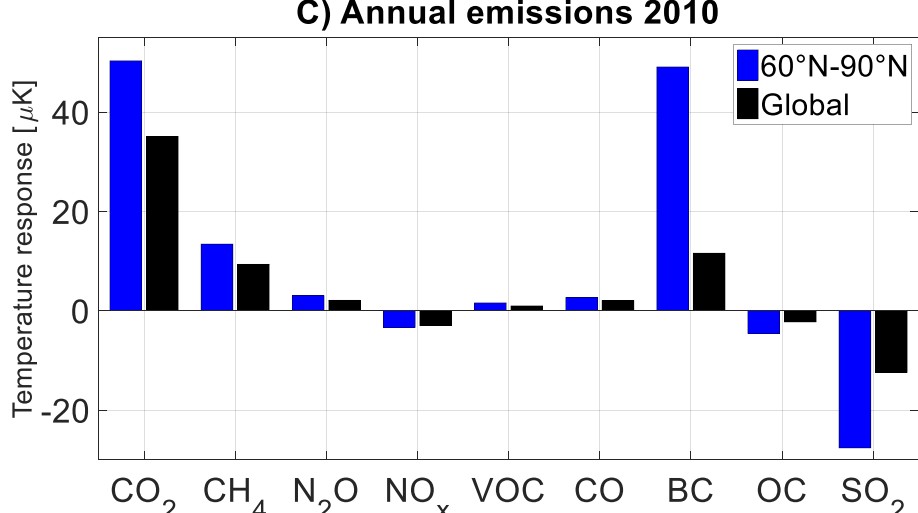

**Figure 8: Global and Arctic (60-90° N) temperature responses (µK) to Finnish emissions based on ARTP values in Aamaas et al. (2017). As for most of the figures, the temperature response is the mean response 1 to 25 years after a pulse emission.**