# Peer review of "Climate Impact of Finnish Air Pollutants and Greenhouse Gases using Multiple Emission Metrics"

_Atmospheric Chemistry and Physics, 2018_

## Referee Comment (RC1) · Collins (Referee) · 3 Jan 2019

This is a straightforward application of the climate metrics of Aamaas et al. 2016 and 2017 to emissions from Finland. The methodology is appropriate so the work is acceptable to publish after minor corrections, but it would have been useful to have some more interesting conclusions.

The discussion about ARTP vs AGTP is confused, particularly since it is not clear whether this is due to latitudinal efficacies or different treatment of BC on snow. Is this discussion really necessary at all? Presumably the Aamaas et al. 2017 are the preferred metrics – so maybe stick with them to simplify the discussion?

[Figure]

If it is going to be included the comparison with Sand et al. 2016 needs to be done better. In their figure 2 they show very much higher temperature response per unit emission for the Nordic Countries compared to Europe. It is surprising therefore that the BC response in this paper (figure 7) is so similar to that using Sand et al. It would seem much more appropriate to use Sand et al. to get the scaling between Nordic and Europe and apply this scaling to the values in this paper.

The mean(1-25) metric is not an obvious choice, and the authors have not explained how it fits in with any nationally or internationally agreed policies. Shindell et al. 2017 indeed admit: "We chose the mean temperature (rather than end-point temperature) to incentivize early action" rather than for any scientific or policy reason.

Line 32-36: I don't think there is any agreement on the definitions of SLCFs or SLCPs – maybe there should be. UNEP (2011) used SLCF when discussing warming agents, IPCC AR6 WG III report used SLCP to refer to warming agents, IPCC SR1.5 stated that SLCP was an equivalent alternative term to SLCF.

Line 51: "metrics is" should be "metrics are".

Line 166: You are actually assuming the pattern is *exactly* the same for all GHGs.

Line 172: Re-phrase "our main pick".

Line 188: "the original" should be "their original"

Line 195: "combining" should be "convolving"

Lines 200-202: This description is too abrupt – the reader would need to be an expert in ARTPs to follow the argument. It either needs a longer explanation to guide the less expert or removing.

Line 209: The AGTP(1-25) is presumably equal to the iGTP of Peters et al. (2011) divided by the time horizon. This should be mentioned.

Lines 240-245: Some comment on the reasons for using emission pulses rather than

emission steps should be provided here. While the pulse gives the mathematically useful Green's function, the convolution with a step emission could be considered more representative of the climate impact of Finland continuing to emit at 2010 levels.

Lines 258-262: Again, if you really need to compared AGTPs vs ARTPs then this needs to be explained for those readers who haven't read Aamaas et al. 2017.

Line 274: If authors consider the relative importance of BC using AGTP and ARTP to be an important point they need to explain why, otherwise this just seems to be a random fact. Alternatively this sentence can be removed.

Section 3.2.4: This section seems to suggest that the main difference between AGTP and ARTP is not the latitudinal dependence of the efficacy, but the different treatments of BC on snow in Aamaas et al. 2016 and 2017. If so, then this should be made more explicit earlier on.

Lines 356-357: This seems a confusing policy message – why should a Finnish policy maker need to know which metric is being used when implementing wintertime BC control measures in Finland?

Lines 392: Why do Sand et al. 2016 have a much larger indirect effect? Is it due to a different model?

Section 4: This section needs to be structured. There is no obvious story being told here. I suggest having the Discussions and Conclusions separately so the Conclusions can be more tightly written in such a way that the reader is clear what the key messages are they should take from this paper.

Lines 445-449: This justification of the mean(1-25) metric is very weak. The argument seems to be purely that Shindell et al. 2017 suggested it.

Lines 481-483: The mean(1-25) metric wasn't "evaluated to be useful" in Shindell et al. 2017, it was simply devised "to incentivize early action".

Lines 483-484: Surely the appropriate metrics for Arctic warming by 2040 or 2050 should be endpoint metrics for 21 or 31 years (e.g. for a start in 2019) rather than a mean over that period.

Figure 7: This needs labelling on the figure to distinguish the Aamaas and Sand values.

[Figure]

---

## Referee Comment (RC2) · Anonymous Referee #2 · 9 Jan 2019

This is an application on known metrics to the context of Finland which is useful. Although there is nothing new or unexpected in the overall results presented in the paper, the findings provide useful insights on the roles and contributions of the various greenhouse gases and SLCFs, as well as the different economic sectors on climate change in Finland and the Arctic region. The methodology used in this work can be replicated and adapted to other countries and regions which makes this work useful.

The analysis focused mainly on Global Warming Potential (GWP), the Global Temperature change Potential (GTP) and the Regional Temperature change Potential (RTP) metrics. However, new metrics have been proposed recently, and the authors

should have considered including these new metrics in this work to increase the overall robustness of the findings, or at the minimum, the authors should have provided justification on why they did not consider these new metrics relevant to this present work. See for example, (1) https://www.nature.com/articles/s41612-018-0026-8 (2) https://www.nature.com/articles/nclimate2998 (3) https://www.oxfordmartin.ox.ac.uk/publications/view/2601 for these recently proposed metrics

---

## Author Comment (AC1) · 22 Mar 2019

RC1:

RC1/1 Comment from the referee: The discussion about ARTP vs AGTP is confused, particularly since it is not clear whether this is due to latitudinal efficacies or different treatment of BC on snow. Is this discussion really necessary at all? Presumably the Aamaas et al. 2017 are the preferred metrics – so maybe stick with them to simplify the discussion?

If it is going to be included, the comparison with Sand et al. 2016 needs to be done

better. In their figure 2 they show very much higher temperature response per unit emission for the Nordic Countries compared to Europe. It is surprising therefore that the BC response in this paper (figure 7) is so similar to that using Sand et al. It would seem much more appropriate to use Sand et al. to get the scaling between Nordic and Europe and apply this scaling to the values in this paper.

Author's response: Comment about ARTP vs AGTP - We agree with the reviewer that the Aamaas et al. (2017) paper has the preferred metrics for our case, and that we base most of our findings on that study. We have reduced the content on AGTP and differences between AGTP and ARTP. AGTP coefficients from Aamaas et al. (2016) are now only used in Figure 2 where we compare GWPs and GTPs. Comment on scaling ARTP values with Sand et al. paper: We thank the reviewer for this good idea about scaling the Arctic responses using the Sand et al. (2016) paper. We did consider this approach in our initial work, but declined as the Sand et al. (2016) paper does not give complete datasets. We have been in contact with the first author to get out as much coefficients as possible. The limitations of the Sand et al. (2016) paper are that they only provide coefficients for ARTP calculations for the Arctic latitude band, they only provide coefficients for BC, OC, and SO2, not for the ozone precursors, and the emissions regions are also not matching completely with ours. However, after looking at this issue again, we agree with the reviewer that the scaling using Sand et al. (2016) paper is an improvement. We are only able to do this scaling for the Arctic. For the ozone precursors and NH3, this scaling must be based on assumptions and simplifications. As ozone is grouped together by Sand et al. (2016), we have used this averaged scaling for NOx, VOC, and OC. This scaling is 1.00, which also convinces us that the scaling approach is also ok for the ozone precursors. For NH3, we have used the average for BC, OC, and SO2 as a scaling factor. Author's changes in manuscript: This comment has resulted in a number of changes in the manuscript. We have reduced the discussion on comparing Aamaas et al. (2017) and Sand et al. (2016) significantly, deleted Figure 7, as well as updated Figures 3-6 and 8. This scaling increases the ARTP values for the aerosols somewhat. We have updated all

discussions, numbers, values in the Tables, given the slightly revised findings.

RC1/2 Comment from the referee: The mean(1-25) metric is not an obvious choice, and the authors have not explained how it fits in with any nationally or internationally agreed policies. Shindell et al. 2017 indeed admit: "We chose the mean temperature (rather than end-point temperature) to incentivize early action" rather than for any scientific or policy reason.

Author's response: The referee is right, we did not have any firm scientific basis to rely on the mean(1-25) metric. As air pollutants (SLCPs) were a key element in our work we wanted to have more emphasis on near-term climate change as pointed out in the last paragraph of section 2.2. As pointed out by Shindell et al. 2017 a mean metric, compared with an end-point metric, gives more weight to shorter lived species and thus incentivizes early action on for example BC. Author's changes in manuscript: We have modified the last paragraph of Section 2.2: "Our climate impact dataset can be analyzed in many different dimensions, such as for different time scales, for different emission sectors, for different processes, for pulse or scenario emissions. We show some examples. As we focus on near term climate change and the global and regional temperature, most of the discussion in this paper utilizes ARTP for the mean warming in the first 25 years after a pulse emission, as recently proposed by Shindell et al. (2017). Mean(ARTP(1-25)) is the average temperature response over the time period, which differ from ARTP(25) being a snapshot at the time horizon of 25 years. We want to point out that our choice of metric is not based on a thorough scientific analysis, but rather a subjective choice to study in more detail the near-term climate impacts and the importance of short-lived species. To balance the choice we compare it with some other known climate metrics."

RC1/3 Comment from the referee: Line 32-36: I don't think there is any agreement on the definitions of SLCFs or SLCPs – maybe there should be. UNEP (2011) used SLCF when discussing warming agents, IPCC AR6 WG III report used SLCP to refer to warming agents, IPCC SR1.5 stated that SLCP was an equivalent alternative term

to SLCF.

Author's response: We agree with the referee that there is no universal agreement on the use of the terms SLCFs or SLCPs. We have highlighted this in the text now. We chose to use the terms as in the IPCC SR1.5 (AnnexI, Glossary). Author's changes in manuscript: We have replaced the end of the first paragraph of Introduction with "In this study we use the terms as in the IPCC Global Warming of 1.5°C Special Report (IPCC 2018) where: (1) SLCFs refer to both cooling and warming species and include methane ($CH_4$), ozone ($O_3$) and aerosols (i.e., black carbon BC, organic carbon OC and sulfate), or their precursors, as well as some halogenated species; (2) SLCPs refer only to the warming SLCFs. Policies focusing on SLCPs have been suggested as supplements to greenhouse gas reductions (UNEP/WMO 2011, Shindell et al. 2012, Rogelj et al. 2014, Stohl et al 2015, Shindell et al. 2017)." Also we added the reference of the IPCC SR1.5 to the list of references: "IPCC, 2018: Annex I: Glossary [Matthews, J.B.R. (ed.)]. In: Global Warming of 1.5°C. An IPCC Special Report on the impacts of global warming of 1.5°C above pre-industrial levels and related global greenhouse gas emission pathways, in the context of strengthening the global response to the threat of climate change, sustainable development, and efforts to eradicate poverty [Masson-Delmotte, V., P. Zhai, H.-O. Pörtner, D. Roberts, J. Skea, P.R. Shukla, A. Pirani, W. Moufouma-Okia, C. Péan, R. Pidcock, S. Connors, J.B.R. Matthews, Y. Chen, X. Zhou, M.I. Gomis, E. Lonnoy, T. Maycock, M. Tignor, and T. Waterfield (eds.)]. In Press"

RC1/4 Comment from the referee: Line 51: "metrics is" should be "metrics are". Author's response: Thank you for noting. Author's changes in the manuscript: We have made the suggested correction.

RC1/5 Comment from the referee: Line 166: You are actually assuming the pattern is *exactly* the same for all GHGs. Author's response: Thank you for noting. Author's changes in the manuscript: We have removed the term "roughly".

RC1/6 Comment from the referee: Line 172: Re-phrase "our main pick". Author's

response: Thank you for noting. Author's changes in the manuscript: We have changed "our main pick" to "we use"

RC1/7 Comment from the referee: Line 188: "the original" should be "their original" Author's response: Thank you for noting. Author's changes in the manuscript: We have made the suggested correction.

RC1/8 Comment from the referee: Line 195: "combining" should be "convolving" Author's response: Thank you for noting. Author's changes in the manuscript: We have made the suggested correction.

RC1/9 Comment from the referee: Lines 200-202: This description is too abrupt – the reader would need to be an expert in ARTPs to follow the argument. It either needs a longer explanation to guide the less expert or removing. Author's response: The RTP concept is explained earlier in Section 2.2, but we agree that a longer explanation would make the case more understandable. Author's changes in the manuscript: 2nd last paragraph of Section 2.2: "As noted, the ARTP method divides the world into four latitude bands, and thus the global temperature response can also be estimated by using the ARTPs and taking the area-weighted global mean basing on the results for the latitude bands."

RC1/10 Comment from the referee: Line 209: The AGTP(1-25) is presumably equal to the iGTP of Peters et al. (2011) divided by the time horizon. This should be mentioned. Author's response: Yes, this is the case. Author's changes in the manuscript: We have added the sentence: "It has similarities with the iGTP concept introduced by Peters et al. (2011)." to the last paragraph of Section 2.2. Peters, G. P., Aamaas, B., Berntsen, T. and Fuglestvedt, J. S. 2011. The integrated global temperature change potential (iGTP) and relationships between emission metrics. Environmental Research Letters 6.

RC1/11 Comment from the referee: Lines 240-245: Some comment on the reasons for using emission pulses rather than emission steps should be provided here. While

the pulse gives the mathematically useful Green's function, the convolution with a step emission could be considered more representative of the climate impact of Finland continuing to emit at 2010 levels. Author's response: We decided to present the emission pulse figures in the beginning of the results section, as they are probably more familiar to many. Also the pulse emissions can give useful information for those working with emission reductions as they can study and compare the effect of emissions of individual years pointing out the sectors were the development has happened or more efforts could be considered. The figures also demonstrate the differences and similarities between the results obtained with the studied metrics, including the GWP100, and point out the particularly the different emphasis given for the SLCPs with the different metrics. We agree that convolution is more representative for continuous emissions as presented in the study. Therefore in section 3.2.2, where we analyze the scenario, we move away from simple pulse considerations and use convolution of pulses for the emission scenarios (Fig. 4). Author's changes in the manuscript: We reworded the start of Section 3.2.2: "While most of our study focuses on emission pulses, we will in this section discuss global temperature responses given a convolution of a Finnish emission scenario and ARTP values." The convolution method is also mentioned in the Methods section. We have also added to the abstract: "We consider both emission pulses and emission scenarios."

RC1/12 Comment from the referee: Lines 258-262: Again, if you really need to compared AGTPs vs ARTPs then this needs to be explained for those readers who haven't read Aamaas et al. 2017. Author's response: We agree (see also our replies to referee comment RC1/9). Author's changes in the manuscript: We have added the sentence: "The ARTP method divides the world into four latitude bands, and the global temperature response is estimated by taking the area-weighted global mean basing on the results for the latitude bands." RC1/13 Comment from the referee: Line 274: If authors consider the relative importance of BC using AGTP and ARTP to be an important point they need to explain why, otherwise this just seems to be a random fact. Alternatively this sentence can be removed. Author's response: The sentence seemed to

miss context. See also reply to RC1/1. Author's changes in the manuscript: We have removed the sentence. RC1/14 Comment from the referee: Section 3.2.4: This section seems to suggest that the main difference between AGTP and ARTP is not the latitudinal dependence of the efficacy, but the different treatments of BC on snow in Aamaas et al. 2016 and 2017. If so, then this should be made more explicit earlier on. Author's response: We agree with the reviewer that a discussion of AGTP and ARTP is unnecessary, and we have removed most of the comparison between AGTP and ARTP. See also our reply to RC1/1. Author's changes in the manuscript: Section 3.2.4 has been shortened, and it now focuses on comparing the results obtained for different seasons. RC1/15 Comment from the referee: Lines 356-357: This seems a confusing policy message – why should a Finnish policy maker need to know which metric is being used when implementing wintertime BC control measures in Finland? Author's response: Both metrics point out to the same conclusion: "From a mitigation perspective, these estimates indicate that attention should be placed on reducing winter emissions of BC." Author's changes in the manuscript: We have modified the sentence accordingly. RC1/16 Comment from the referee: Lines 392: Why do Sand et al. 2016 have a much larger indirect effect? Is it due to a different model? Author's response: We have removed this paragraph as a response to the comment RC1/1 about using Sand et al. (2016) for scaling. We considered a discussion of the indirect effects in Sand et al. (2016) now unnecessary, as we are only interested in the ratios between European emissions and Nordic emissions.

Author's changes in the manuscript: We have removed this paragraph as a response to RC1/1 on using Sand et al. (2016) as scaling.

RC1/17 Comment from the referee: Section 4: This section needs to be structured. There is no obvious story being told here. I suggest having the Discussions and Conclusions separately so the Conclusions can be more tightly written in such a way that the reader is clear what the key messages are they should take from this paper. Author's response: We agree that this section is long and conclusions are not evident from

the discussion. We have followed the referee's suggestion and have written a separate Conclusions chapter that highlights our key conclusions from the work. Also we have reduced content in the Discussion sections. In drafting the conclusions we have concentrated on those that we think might be of interest for a more general audience rather than those interested in the Finnish situation.

Author's changes in the manuscript: We have renamed section 4 as "Discussion" and added a section 5 "Conclusions". We have reduced the content of Section 4.

RC1/18 Comment from the referee: Lines 445-449: This justification of the mean(1-25) metric is very weak. The argument seems to be purely that Shindell et al. 2017 suggested it. Author's response: That is correct. See also reply to Referee comment R1/2. This is a subjective choice to study in more detail the near-term climate impacts and the importance of short-lived species. Author's changes in the manuscript: Sentence "This is a subjective choice to study in more detail the near-term climate impacts and the importance of short-lived species." added to the paragraph.

RC1/19 Comment from the referee: Lines 481-483: The mean(1-25) metric wasn't "evaluated to be useful" in Shindell et al. 2017, it was simply devised "to incentivize early action". Author's response: We agree. Author's changes in the manuscript: We have removed the part "evaluated to be useful" from the sentence.

RC1/20 Comment from the referee: Lines 483-484: Surely the appropriate metrics for Arctic warming by 2040 or 2050 should be endpoint metrics for 21 or 31 years (e.g. for a start in 2019) rather than a mean over that period. Author's response: We agree that this wording does not reflect mean(1-25) metric. We have reworded the sentence Author's changes in the manuscript: The last part of the sentence is changed to: "...from today and until 2040 or 2050."

RC1/21 Comment from the referee: Figure 7: This needs labelling on the figure to distinguish the Aamaas and Sand values. Author's response: We have decided to delete the Figure Author's changes in the manuscript: Figure 7 has been deleted as a

response to RC1/1 on the Sand et al. (2016) scaling. We have included new numbering for all Figures.

RC2: RC2/1 Comment from the referee: The analysis focused mainly on Global Warming Potential (GWP), the Global Temperature change Potential (GTP) and the Regional Temperature change Potential (RTP) metrics. However, new metrics have been proposed recently, and the authors should have considered including these new metrics in this work to increase the overall robustness of the findings, or at the minimum, the authors should have provided justification on why they did not consider these new metrics relevant to this present work. See for example, (1) https://www.nature.com/articles/s41612-018-0026-8 (2) https://www.nature.com/articles/nclimate2998 (3) https://www.oxfordmartin.ox.ac.uk/publications/view/2601 for these recently proposed metrics

Author's response: Thank you for pointing out these recent important and interesting papers. The new usage of the GWP metric seems interesting. We have included an analysis of the Finnish emissions in the period 2000-2030 to the paper (see new Figure 3 and corresponding text).

Author's changes in the manuscript: We have added a new figure (new Figure 3) that compares SLCFs with emissions of $CO_2$ and $N_2O$ in the period 2000-2030 with the metric GWP*(100) as well as corresponding text. We have also introduced this metric in Section 2 Methodology.